# Information overload and parental perspectives on information provided to parents/carers of paediatric patients undergoing elective surgical procedures

Aine Sommerfield[1,2,3,4], David Sommerfield[1,2,3,4], Kenneth Lee[5], Daisy Evans[1,2,3,6], Megan Dodd[1,2], Emily Bell[1,2], Simone Gonsalves[1,2], R. Nazim Khan[3,7], Britta S. von Ungern-Sternberg[1,2,3,4]*

1 Perioperative Medicine Team Telethon Kids Institute, Nedlands, Western Australia, Australia, 2 Department of Anesthesia and Pain Medicine, Perth Children's Hospital, Nedlands, Western Australia, Australia, 3 Institute for Paediatric Perioperative Excellence, The University of Western Australia, Perth, Western Australia, Australia, 4 Division of Emergency Medicine, Anesthesia and Pain Medicine, Medical School, The University of Western Australia, Perth, Western Australia, Australia, 5 Discipline of Pharmacy, School of Allied Health, The University of Western Australia, Perth, Western Australia, Australia, 6 School of Physics, Mathematics and Computing, The University of Western Australia, Perth, Western Australia, Australia, 7 School of Mathematics and Statistics, The University of Western Australia, Perth, Western Australia, Australia

* Britta.regli-vonungern@health.wa.gov.au

**Data Availability Statement:** There are ethical restrictions on sharing the underlying data because the data contains potentially sensitive information.

## Abstract

When parents are expected to play a significant role in the management of their children's health perioperatively, information overload for parents could have particularly detrimental consequences. Our study investigated information communication and overload in 380 parents of children undergoing any elective surgical procedure at our institution. Participants completed an online questionnaire and were asked to respond to a newly designed Information Overload scale based on a modified 5 item Cancer Information Overload Scale and an 8-item atrial fibrillation information overload scale. Nineteen respondents (5%) identified as Aboriginal or Torres Strait Islander. More than a quarter of respondents (n = 102, 27%) primarily spoke a language other than English at home. 56% of respondents (n = 211) indicated that neither themselves nor any of their children had undergone a similar surgery in the past. Most respondents disagreed or strongly disagreed with the majority of the 5-item Information Overload scale statements. University undergraduates had lower total information overload score on average (-1.63, p = 0.002). People who spoke a language other than English had higher total score on average (0.98, p<0.001). Semi-structured qualitative interviews including the BRIEF health literacy screening tool were conducted with 24 parents. 23 interviewees scored 17–20 points in the BRIEF health literacy screening tool, indicating they were able to read and comprehend patient education materials. Overall, parents were satisfied with the amount of information that they received. Very rarely did any parent complain of 'information overload'. Our results show that parents of children undergoing elective paediatric surgery are not suffering from information overload in general, but they do want more information on immediate and late postoperative recovery.

These restrictions are imposed by the Child & Adolescent Health Service who approved the study. Contact information for the Child & Adolescent Health Service to which data requests may be sent: Child and Adolescent Health Service Human Research Ethics Committee 15 Hospital Avenue Nedlands Western Australia 6009 Australia. CAHS. Ethics@health.wa.gov.au +61 08 6456 8639.

**Funding:** BVuS is part funded by the Stan Perron Charitable Foundation and is a recipient of a NHMRC Investigator Grant (2009322). The funders had no role in study design, data collection and analysis, decision to publish, or preparation of the manuscript.

**Competing interests:** The authors have declared that no competing interests exist.

## Introduction

Health literacy includes people's knowledge, motivation and competences to access, understand, appraise, and apply health information to make judgments and take decisions concerning healthcare, disease prevention and health promotion [1]. Evidence from the United States of America [2] and Europe [3] suggest that some patients may have inadequate health literacy levels to self-manage their health conditions; this applies both broadly, and specifically to diseases such as heart failure [4], chronic pain [5] and chronic kidney disease [6].

The consequences of limited health literacy can be further compounded by information overload. In the current information age, patients and families may be exposed to a potentially bewildering array of information sources of varying quality and trustworthiness [7, 8]. As outlined in Obamiro and Lee [9], this information may lead individuals to not paying attention to vital information, incorrectly processing, or even avoiding information, all of which could have a negative impact on health outcomes. Additionally, a recent extensive study assessing consumer priorities for perioperative medicine found that improved communication with parents rated eighth in their top ten priorities [10].

In response to survey data that suggested three-quarters of adults are overwhelmed by cancer information, Jensen et al. [11] developed and validated an 8-item Cancer Information Overload (CIO) scale to evaluate the associations between cancer information overload and cancer-related behaviours such as cancer screening. They found that patients who reported feeling overloaded with cancer information and who consequently had a higher score on the CIO scale, predicted a lower likelihood of colon cancer screening within an 18-month period. The authors concluded that high CIO undermines health efforts. The authors suggest that healthcare providers and public health professionals could use the CIO scale to develop evidence-based communication strategies to mitigate CIO.

Cancer information overload is the most popular topic in studies of consumer health information overload; however, the scale has been adapted for use in diet information, weight management, intensive care next of kin and coronary heart disease [12]. Obamiro and Lee [9] assessed the validity of the 8-item scale for use with Australian atrial fibrillation patients. The authors also validated a modified 5-item version for the same population [13]. Both the original 8-item and reduced 5-item scales were reported to be valid and reliable for measuring health information overload among adult Australians living with atrial fibrillation [9]. They suggested that both the 8-item and reduced 5-item modified CIO scales may be suitably adapted for health information overload in patients with other chronic diseases; however the reduced 5-item modified CIO scale may be preferable if time is limited to collect participant responses or in attempts to mitigate respondent fatigue and improve response quality.

Information overload may apply not just to chronic conditions. It has been suggested that information overload could negatively impact patients' abilities to contribute to their healthcare management [12]. In the paediatric setting, where parents are pivotal in making decisions and managing their children's health perioperatively, the impact of information overload on parents remains largely unexplored, despite its potentially significant consequences. Parents attending hospital with their children for surgery see an array of health professionals perioperatively who provide information on their child's condition, treatment, and ongoing management [10, 14, 15]. Additionally, the perioperative period is notably stressful for parents, with heightened levels of anxiety for both parents and child, further burdening the mental load of parents. Therefore, our research team conducted a study among parents and carers of children undergoing any elective surgical procedure at our institution with the objective to investigate information communication and overload.

## Materials and methods

This study was approved by the Child and Adolescent Health Service Human Research Ethics Committee (RGS000003994) and recognised by The University of Western Australia Human Research Ethics Committee (2021/ET000512).

The objectives were to:

1. Assess the usefulness of a modified version of the Cancer Information Overload scale with parents/carers of paediatric patients undergoing any elective surgical procedure;

2. Identify the level of information overload, if any, among such parents/carers;

3. Explore information communication and overload from the parents' /carers' perspectives.

### Survey of participants on day of surgery

Participants (parents of children undergoing any elective surgical procedure) were invited to participate through poster in the perioperative area advertising with QR code and/or by members of the research team. A questionnaire was developed in Qualtrics (Utah, USA,) and disseminated to participants from May 2021 to October 2021. At the start of the survey parents were asked to consent to participate in the survey. Demographic information (age, gender, Aboriginal or Torres Strait Islander self-identification, postal code, annual income, highest educational level, and employment status) was recorded by respondents. Participants were also asked how many health professionals their child had seen prior to the current surgery in relation to the surgery/condition, e.g., general practitioner, specialist, child health nurse or dentist. Postcodes were classified as metropolitan (metro), regional, or remote based on the Australian Bureau of Statistics Statistical Geography Standard Correspondences Postcode 2017 to Remoteness Area 2016 [16]. Postcodes classified as being in "Major Cities of Australia" were grouped under "metro", postcodes classified as "Inner Regional Australia" or "Outer Regional Australia" were grouped under "regional", and those classified as "Remote Australia" or "Very Remote Australia" were grouped under "remote".

Participants were asked to respond to an Information Overload scale. The original 8-item CIO scale contained four points per item (from strongly disagree with a value of 1, to strongly agree with a value of 4) [11]. Scoring of the original CIO scale was performed by summing the value of each item (scores range from 8 to 32), with higher scores indicating a greater degree of health information overload. For this study, we adapted the 5-item CIO scale (scores range from 5 to 20) and the 8-item atrial fibrillation scale [9] using"caring for your children after surgery" (see Table 1).

Given that there is no previous data to enable sample size estimation based on power, we were unable to perform a power calculation. As such, we used conservative estimates, using a

**Table 1. Modified 5-item information overload scale used.**

| Item | Modified IO adapted for Perioperative paediatric environment |
|------|--------------------------------------------------------------|
| 1 | There are so many different recommendations about caring for your child after surgery, it's hard to know which ones to follow |
| 2 | It has gotten to the point where I don't even care to hear new information about caring for my child after surgery |
| 3 | Information about caring for my child after surgery all starts to sound the same after a while |
| 4 | I forget most of the information about caring for my child after surgery right after I hear it |
| 5 | I feel overloaded by the amount of information about caring for my child after surgery I am supposed to know |

proportion of 0.5 for parental information overload, an unknown large population size, a 95% level of confidence and a margin of error of 5%, resulting in a minimum sample size of 384 [17].

## Semi-structured qualitative interviews

To provide complementary data to the survey, semi-structured qualitative interviews were conducted with a subgroup of parents of patients who had undergone elective surgery and who indicated they would be open to further contact in the survey. Prior to the interviews, the participants gave informed consent to participate in the study. A convenience sample were contacted by telephone by research assistants to organise interviews. When staff was available, all potentially eligible families were contacted. An Interview Guide (S1 File) was used to ensure consistency between interviews; however, questions remained open-ended, allowing parents to explore themes they deemed important. The Interview Guide began with an introduction to the research team, details of what the study involved and outlined the interview process. The targeted number of participants was 20, as this is typically large enough for data saturation to occur [18]. For the present study, data saturation was defined as the point where no new themes could be independently identified by two or more researchers from the research team. The interviews were undertaken by trained, experienced members of the research staff over the phone or via video call at a time that was convenient for the interviewee. Only the interviewee and interviewer were present during the interviews. No repeat interviews were conducted. Field notes were taken by the interviewers, in order to highlight preliminary themes and identify when data saturation had been reached. Participants' responses were recorded via an audio recorder and transcribed verbatim. An abbreviated health literacy assessment was conducted using the validated 4-item BRIEF health literacy screening tool to provide further demographic context [19]. The demographic and health literacy data collected during the interview were analysed to provide context on the participant group.

## Data analysis

Analysis of survey responses was conducted using R Version 4.2 [20]. Responses to individual items in the modified 5-item information overload scale were described using medians, lower quartiles (LQ) and upper quartiles (UQ). To establish evidence for structural validity, construct validity and reliability, psychometric properties of the scale were explored, with internal consistency assessed by considering Cronbach's alpha, and the factor structure of the scale established by performing exploratory factor analysis (EFA) with oblique rotation.

To identify variables associated with information overload, multiple linear regression analysis was performed for the modified 5-item information overload scale total score. Variables considered in the regression analysis included participants' demographic characteristics (gender, age group, Aboriginal or Torres Strait Islander self-identification, whether a language other than English is spoken at home, education level, employment status, income, postcode remoteness) and whether or not the respondent or any of their children had previously had a similar surgery. Interaction terms between variables were included based on exploration of interaction plots and cross-tables. Backwards stepwise regression was used to reduce the model to significant covariates, where statistical significance was taken at $p < 0.05$. Model assumptions were verified by inspection of diagnostic plots. The summed score of multiple Likert type items into a Likert scale score can be treated as interval data, and is therefore suitable for analysis by linear regression [21]. As lower education level has previously been found to be significantly associated with greater cancer information overload [9], we predicted that education levels would be negatively correlated with our modified information overload scale

scores. While there are few studies examining the relationship between other demographic characteristics and health information overload, we predicted there will be no significant correlations with our other demographic characteristics (age, gender, postcode, annual income, and employment status). Given the low rate of missing data, regression analysis was performed on complete cases only.

Thematic analysis was undertaken on the interview transcripts, using the qualitative data analysis software NVivo® 11. Researchers utilized the Triangulation Framework method of analysis to identify themes [22]. The Framework method was chosen because it is useful for research teams comprising of multiple analysts with varying levels of qualitative research experience, as it provides a structured approach to qualitative analysis [23]. Descriptive (quantitative) statistics of demographic and health literacy information were performed using Microsoft Excel®.

The proposed study was guided by various methods of trustworthiness proposed by Lincoln and Guba [23]. Analyst triangulation was used during the coding process to improve the reliability of the data, acknowledging that each researcher brings their own attitudes, values and world views to their interpretation of the interview transcripts. Incorporating multiple researcher viewpoints and skillsets into the analysis mitigates the risk of bias or favouring one particular point of view. Prior to analysis, each researcher also reflected on their pre-existing perspectives, assumptions and expectations of the data in order to identify implicit biases, thus further improving the reliability of the data.

Three researchers (EB, MD and SG) firstly read through all interview transcripts to familiarise themselves with the data. One initial transcript was coded by three researchers (EB, MD and SG) and principal investigator KL. Researchers then met to discuss discrepancies in coding and reconcile any differences. Following this discussion, a working analytical framework was developed in consultation with KL and the study investigators. This framework comprised of a table of codes identified from the data with associated definitions. Researchers EB, MD and SG then coded the remaining transcripts based on this framework, adding new codes and definitions as they arose in discussion with the study team. Coded data was combined into a framework matrix showing participant data for each code, including direct and/or summarised participant quotes. Codes were then tentatively grouped together according to emerging themes. Final themes were established by mapping connections within and between participant responses in collaboration with the research team.

Results are reported in accordance with COREQ (Consolidated criteria for reporting qualitative research) [24] or STROBE guidelines [25] as appropriate.

## Results

### Survey of participants on day of surgery

A total of 382 respondents consented to the Day of Surgery survey. Of these, 2 participants did not respond to any of the questions following consent, and were subsequently excluded from the analysis, leaving 380 participants (female = 290, 76%) from 126 postcodes. The postcode responses indicated that 304 respondents (80%) were in metropolitan areas, with 71 from regional/remote areas and 5 missing.

Table 2 shows a summary of the participant characteristics. Nineteen respondents (5%) identified as Aboriginal or Torres Strait Islander. More than a quarter of respondents (n = 102, 27%) primarily spoke a language other than English at home. 56% of respondents (n = 211) indicated that neither themselves nor any of their children had undergone a similar surgery in the past, and 53% (n = 202) indicated that neither themselves nor any of their children had undergone multiple surgeries in the past.

**Table 2. Demographic details of day of surgery survey respondents.**

|  | N | % |
|---|---|---|
| Gender, female | 290 | 76 |
| Aboriginal or Torres Strait Islander | 19 | 5 |
| Language other than English spoken at home | 102 | 27 |
| Parental Age group |  |  |
| 18–24 | 5 | 1 |
| 25–34 | 108 | 28 |
| 35–44 | 173 | 46 |
| 45–54 | 84 | 22 |
| 55 and over | 10 | 3 |
| Number of health professionals seen |  |  |
| One | 45 | 12 |
| Two-five | 246 | 65 |
| More than five | 89 | 23 |
| Annual household income |  |  |
| Less than $25,000 | 21 | 6 |
| $25,000-$75,000 | 94 | 25 |
| $76,000-$125,000 | 94 | 25 |
| $126,000-$250,000 | 94 | 25 |
| Over $250,000 | 25 | 7 |
| Prefer not to say | 52 | 14 |
| Highest education level |  |  |
| Did not complete high school | 39 | 10 |
| High school completion | 65 | 17 |
| Technical/TAFE qualification | 133 | 35 |
| Undergraduate university degree | 74 | 19 |
| Postgraduate university degree | 69 | 18 |
| Home postcode |  |  |
| Metro | 304 | 80 |
| Regional | 55 | 14 |
| Remote | 16 | 4 |
| NA | 5 | 1 |
| Employment status |  |  |
| Not currently employed, not looking | 61 | 16 |
| Not currently employed, looking | 18 | 5 |
| Employed, part time hours | 128 | 34 |
| Employed, full time hours | 122 | 32 |
| Self-employed/business owner | 51 | 13 |

## Information overload

The responses to the individual items from the modified 5-item information overload scale are summarised in Table 3. Most respondents disagreed or strongly disagreed with the majority of the 5 item statements. Item 3 ("Information about caring for my child after surgery all starts to sound the same after a while") had the highest rate of agreement, with 29% agreeing or strongly agreeing with the statement. A score of 2 ("disagree") was most commonly selected for all items except for item 2 ("It has gotten to the point where I don't even care to hear new information about caring for my child after surgery"), for which a score of 1 ("strongly disagree") was most commonly selected. The mean (SD) of the total score was 9.2 (2.5), and

the median (LQ, UQ) was 10 (7, 11). The maximum total score was 18, and the minimum was 5.

Cronbach's alpha for the modified 5-item information overload scale was 0.78, indicating acceptable internal consistency, evidence for acceptable reliability [26]. Exploration of the factor structure indicated a one- or two-factor solution; however, the individual modified information overload scale items were summed to a one-factor solution to align with the design of the original score. Details of the results of EFA are given in S2 File.

### Regression analysis

Given the five participants with missing postcode, the complete-case multiple regression analysis was performed on the remaining 375 participants. Unadjusted regression estimates from single-variable models for total score on each candidate independent variable are reported in Table 1 in S3 File. However, these do not reflect the true effect of each variable as this analysis does not account for the effect of confounding variables. Interaction terms for education with Aboriginal or Torres Strait Islander, age group with education, and language other than English (LOTE) with education were included in the initial model based on inspection of interaction plots (Figs 1 to 3 in S3 File). Parameter estimates from the full multivariable model before reduction of covariates are shown in Table 2 in S3 File. Following backwards stepwise variable selection, the estimated coefficients and 95% CI from the multiple linear regression for the total modified 5-item information overload scale scores are shown in Table 4. Model diagnostics confirmed that the normal linear model assumptions were verified. People who spoke a language other than English (LOTE) had higher total score on average (0.98, p<0.001).

Education level was shown to have an effect on information overload. None of the Indigenous Australian participants in this cohort reported having university degrees; however, those who completed high school had higher mean information overload total scores compared to those who had technical/TAFE education or who had not completed high school. Of respondents who were not Aboriginal or Torres Strait Islander, those who had TAFE or university qualifications had lower average information overload total score compared to those who had not completed high school (TAFE -1.21, p = 0.013; undergraduate -1.63, p = 0.002; postgraduate -1.12, p = 0.031), indicating lower information overload.

The interaction between Aboriginal or Torres Strait Islander heritage and education level was statistically significant (see Fig 1). Aboriginal and Torres Strait Islander people whose highest education was high school completion had significantly higher information overload scores than non-Aboriginal people with the same education level (3.4, p = 0.020). No comparison could be made between Aboriginal or Torres Strait Islander and non-Aboriginal participants with undergraduate or postgraduate university degrees, since none of the Indigenous participants in this cohort reported having university degrees. Fig 1 shows the model-estimated mean score and 95% confidence interval for different levels of education and Aboriginal and/or Torres Strait Islander heritage.

### Semi-structured qualitative interviews

Results are reported in accordance with COREQ (Consolidated criteria for reporting qualitative research) [24]. Interviews and data collection were performed by three researchers with experience in qualitative research (authors EB, SB and MD). The interviewers had no prior relationship with the participants. Participants were aware that the research was about their children's experience with surgery and how they felt about the amount of information they received. A convenience sample of 24 parents took part in semi-structured interviews chosen from respondents who had indicated a willingness to participate in the interview stage (n = 136 of 382; 35; 35.6%). The interviews were conducted between 20 and 59 days following

**Table 3. Median, lower quartile (LQ) and upper quartile (UQ) of cohort response scores to each of the 5 items on the modified information overload scale (code in brackets).** Some percentages may not add to 100 due to rounding.

| Item | Statement | Median (LQ, UQ) | Frequency selected, n (%) | | | |
|------|-----------|-----------------|---------------------------|---|---|---|
| | | | Strongly Disagree (1) | Disagree (2) | Agree (3) | Strongly Agree (4) |
| 1 | There are so many different recommendations about caring for your child after surgery, it's hard to know which ones to follow | 2 (2, 2) | 77 (20%) | 235 (62%) | 58 (15%) | 10 (3%) |
| 2 | It has gotten to the point where I don't even care to hear new information about caring for my child after surgery | 1 (1, 2) | 204 (54%) | 154 (41%) | 17 (4%) | 5 (1%) |
| 3 | Information about caring for my child after surgery all starts to sound the same after a while | 2 (2, 3) | 90 (24%) | 181 (48%) | 102 (27%) | 7 (2%) |
| 4 | I forget most of the information about caring for my child after surgery right after I hear it | 2 (1, 2) | 123 (32%) | 212 (56%) | 42 (11%) | 3 (1%) |
| 5 | I feel overloaded by the amount of information I am supposed to know about caring for my child after surgery | 2 (1, 2) | 119 (31%) | 213 (56%) | 43 (11%) | 5 (1%) |
| | TOTAL SCORE | 10 (7, 11) | | | | |

their child's operation. Of the parents who were contacted for an interview, three parents declined further participation, 28 were interviewed but 4 of these were emergency surgery parents who were subsequently not analysed as the study was for planned elective surgery only. Two interviews were pending when it was determined that data saturation had been reached and therefore these interviews were not scheduled.

The BRIEF health literacy screening tool was used with 23 interviewees scoring 17–20 points, indicating they were able to read and comprehend patient education materials. One interviewee had a marginal score which indicated they may need assistance or may struggle with patient education materials. Thirteen parents reported seeing 2–5 health professionals and nine parents more than five health professionals. Table 5 summarises the characteristics of interview participants.

Theme codes and comments from the qualitative analysis are summarised in Table 6. Overall, parents were satisfied with the amount of information that they received. Very rarely did any parent complain of 'information overload', and of those who did were referring to the

**Table 4. The estimated coefficients and 95% CI from a multiple linear regression for the modified 5-item information overload scale total score.** REF indicates the reference level for that factor.

| Variable | | Estimate | Std. Error | p-value | 95% CI | |
|----------|--|----------|------------|---------|--------|--|
| | | | | | 2.5% | 97.5% |
| **Intercept** | | 9.94 | 0.44 | <0.001 | 9.08 | 10.8 |
| **Aboriginal or Torres Strait Islander** | No | REF | | | | |
| | Yes | -0.05 | 0.95 | 0.958 | -1.92 | 1.82 |
| | Prefer not to say | 0.21 | 2.4 | 0.931 | -4.5 | 4.92 |
| **Education** | Did not complete high school | REF | | | | |
| | High school completion | -0.48 | 0.53 | 0.362 | -1.53 | 0.56 |
| | Technical/TAFE qualification | **-1.21** | **0.48** | **0.013** | **-2.16** | **-0.26** |
| | Undergraduate University degree | **-1.63** | **0.51** | **0.002** | **-2.64** | **-0.62** |
| | Postgraduate university degree | **-1.12** | **0.52** | **0.031** | **-2.15** | **-0.1** |
| **LOTE** | No | REF | | | | |
| | Yes | **0.98** | **0.28** | **<0.001** | **0.42** | **1.53** |
| **Aboriginal or Torres Strait Islander status * Education** | Yes * High school completion | **3.4** | **1.45** | **0.020** | **0.55** | **6.26** |
| | Yes * Technical/TAFE qualification | 0.67 | 1.37 | 0.624 | -2.02 | 3.36 |
| | Prefer not to say * Technical/TAFE qualification | -0.93 | 3.38 | 0.782 | -7.58 | 5.71 |

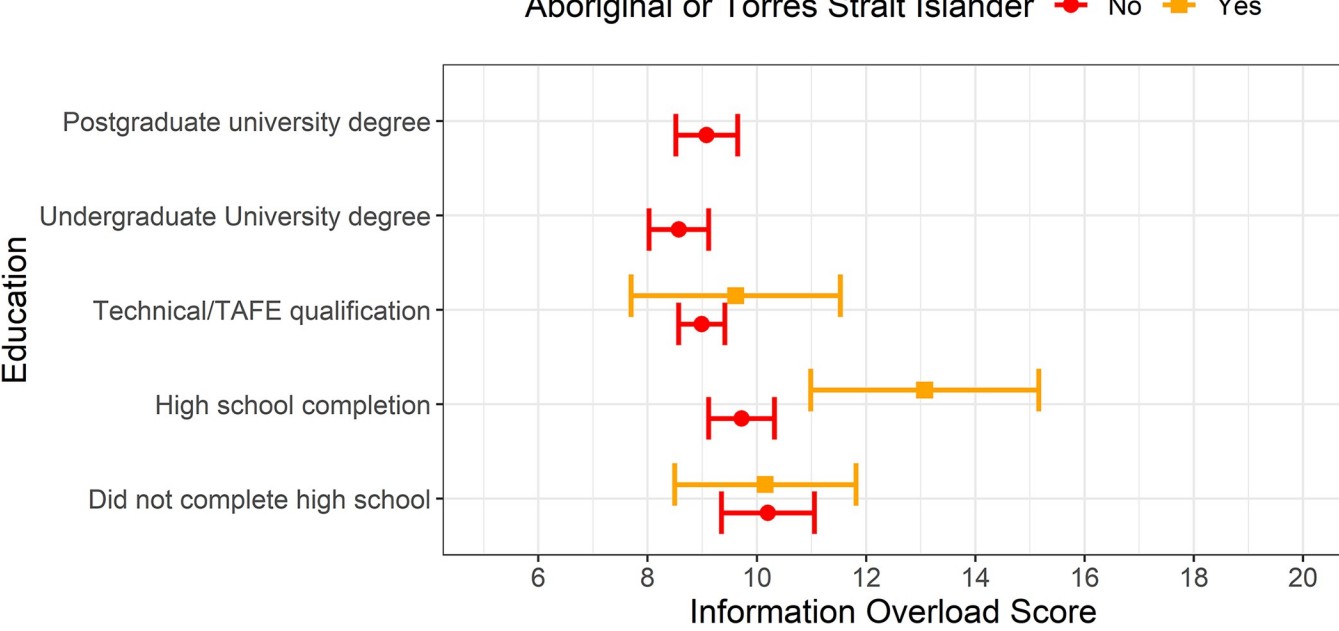

**Fig 1. Model-estimated mean and 95% CI modified 5-item information overload score for different levels of education and Aboriginal or Torres Strait Islander heritage, demonstrating the effect of their interaction on the score.** The level "Prefer not to say" of Aboriginal or Torres Strait Islander is not shown due to small numbers and wide confidence intervals.

overall stress of the hospital environment/staff communication. For example, "*We were dealing with a sick child that was crying so we couldn't concentrate on anything that was told to me, I would have forgotten everything so having it in writing was excellent*" (PB198).

Parents mainly found communication between nurses/doctors and families was positive and effective. However, for some parents, the communication of verbal information by staff was not at the parent's comprehension level, with one parent stating "*The actual information is clear but then it gets distorted through their (staff) communication*" (PB70). Another parent felt the manner of information delivery could be improved "*Bit more sensitivity to the format and delivery, as opposed to the information itself would be a good investment of time*" (PB85).

Parents were pleased with the written information that was given to them to take home. They reported finding this the most effective way to manage the information after surgery. Some parents suggested having more personalised information would be of benefit.

Parents with more experience in the healthcare system (either with themselves or children who had previous surgeries) found it easier to retain the information and had already developed strategies on how to process it. One parent noted, "*I guess unfortunate that I have a lot of experience myself in the hospital, unfortunate that I am already aware of all the information, so it is something that is automatic to me*" (PB70).

Unsurprisingly, parents who themselves were medical professionals (nurses, GP etc.) found the process much less overwhelming. Previous experience in the hospital and previous experience with personal health concerns and/or chronic illness seemed to play a significant role in how parents perceive and process the information given to them.

## Discussion

Most respondents to the initial questionnaire disagreed or strongly disagreed with most of the individual item statements in the modified 5-item information overload scale, indicating that

**Table 5. Demographic characteristics of the parents who took part in stage 3 semi-structured interviews.**

| Variable | | Number (%) |
|---|---|---|
| Aboriginal status or Torres Strait Islander or | Yes | 1 (4.2%) |
| | No | 23 (95.8%) |
| | Prefer not to say | 0 |
| Age group | 18–24 years | |
| | 25–34 | 5 (20.8%) |
| | 35–44 | 14 (58.3%) |
| | 45–54 | 5 (20.8%) |
| | 55 and over | |
| Annual Household Income | <$25,000 | 1(4.2%) |
| | $25000 TO $75000 | 5 (20.8%) |
| | $76000 TO $125000 | 7 (25.9%) |
| | $126000 TO $250000 | 7(25.9%) |
| | >$250000 | 2 (8.3%) |
| | Prefer not to say | 2(8.3%) |
| Employment status | Not currently employed, not looking for a job | 4 (16.7%) |
| | Searching for job | 0 |
| | Employed part time | 8 (33.3%) |
| | Employed full time | 10 (41.7%) |
| | Self-employed /business owner | 1 (4.2%) |
| Gender | Female | 17 (70.85) |
| | Male | 7 (25.9%) |
| | Prefer not to say | 0 |
| Highest Education level achieved | Did not complete high school | 1 (4.2%) |
| | High school completion | 5 (20.8%) |
| | Technical/TAFE | 7 (25.9%) |
| | Undergraduate degree | 6 (25%) |
| | Postgraduate degree | 5 (20.8%) |
| Language other than English spoken at home | Yes | 1 (4.2%) |
| | No | 23 (95.8%) |
| | Prefer not to say | 0 |
| Number of health professionals seen | One | 2(8.3%) |
| | Two-Five | 13 (54.2%) |
| | More than 5 | 9 (37.5%) |
| Postcode | Metro | 16 (66.7%) |
| | Rural | 8 (33.3%) |

they did not feel overloaded with information about the care of their children after surgery. The linear regression analysis revealed that the interaction between education level and Indigenous Australian self-identification was statistically significant. We found a mean difference in information overload between Indigenous and non-Indigenous Australians that depended on their highest education level. Higher education levels corresponded to lower information overload in our cohort. In a 2020 scoping review by Khaleel et al. [12] lower education level was associated with health information overload in several studies they reviewed; however, one study had reported a higher level of education to be associated with diet information overload. This diet study had a high proportion of Hispanic/Latino participants and more than 20% had below high school education level. Additionally, many of the participants received nutrition

**Table 6. The key themes, barriers and enablers surrounding the information received by parents regarding their child's surgery, identified from thematic analysis of semi-structured interview transcripts.**

| Code | Subcode | Number of times raised | Comment |
|---|---|---|---|
| **Barriers to effective communication with parents** | Conflicting Information | N = 5 | ■ Discrepancy between doctor and nurse information resulting in parents being unsure about what information to follow<br>■ Conflicting information regarding surgery times/agenda for the day |
| | Hospital Environment | N = 2 | ■ General stress of the hospital or nature of the procedure made it challenging for parents to retain information<br>■ Parents needing to organise hospital visit around other children and other aspects of their life which made it overwhelming (not the information itself, rather situational factors) |
| | Own research and prior experiences | N = 2 | ■ Parents relying on the internet to research information, which conflicted with hospital information leading to confusion<br>■ This was particularly difficult for parents with their own health challenges (e.g. ADHD), which made it challenging to retain information |
| | Poor Communication | N = 21 | ■ Unclear instructions or vague information given to parents led to confusion<br>■ Lack of information (this was a general complaint by parents, feeling as though they were not given enough information regarding the procedure or postoperative care)<br>■ Communication of information was not at the parent's comprehension level<br>■ Communication not tailored enough to the participants<br>■ Rate of delivery of information was too fast |
| **Enablers of effective communication with parents** | Written Information | N = 31 | ■ Parents like having a written handout to refer to, hospital environment can be distracting at times so a take-home sheet is very useful |
| | Verbal Information | N = 24 | ■ Staff communication was clear and effective<br>■ Staff engaged with the child and made them feel included in the process<br>■ Most people complimented the verbal delivery and liked having written take-home information to refer to |
| | Own Research or Experience | N = 18 | ■ Parents whose children have already had surgery felt far more comfortable, confident, and competent<br>■ Parents who have spent time in the hospital environment have found ways to adapt to information presented<br>■ Parents with medical background (nurse/doctor) found it helpful<br>■ Own research beforehand helped them to feel more comfortable |
| | Rapport and Communication | N = 14 | ■ Staff were open and friendly, asking if the parents feel comfortable which helped parents to retain information |
| **Reported instances of information overload/overwhelm, and factors contributing to it.** | Yes | N = 3 | ■ Overwhelmed when different staff presented different pieces of information<br>■ Overwhelmed when there were instances of conflicting information or miscommunications |
| | No | N = 23 | ■ Most parents did not feel overloaded, they felt they were communicated with well and were comfortable with the amount of information they were given |
| **Suggestions to help facilitate effective communication** | Parental Empowerment | N = 10 | ■ Parents were encouraged to ask questions and feel comfortable to speak up amongst hospital staff |
| | Improved Communication | N = 3 | ■ Ensure staff communication is tailored to the individual and matches the parent's level of understanding |
| | Improved Personalised Information | N = 8 | ■ Parents liked the idea of having more personalised take-home care information, less generic information |
| | Preferred Method | N = 10 | ■ Combination, n = 3, Written n = 3, Multimedia n = 3<br>■ Most parents said it is useful to have written information to take home, found that combination of verbal and written is the most effective<br>■ Some suggested multimedia forms (YouTube, videos etc) |

assistance services. Therefore, the scoping review authors conclude that despite this contradictory finding there is an association between education level and information overload.

Although the proportion of Aboriginal or Torres Strait Islander participants in the online survey cohort mirrored the general population of Western Australia, none of the Indigenous Australian participants in this cohort reported having university degrees; therefore, we cannot make inferences about differences in information overload score between Indigenous and non-Indigenous Australians with university degrees.

We also found that households that spoke a language other than English had higher information overload (0.98, p<0.001). As expected, we found no significant correlations between the total modified 5-item information overload score and the other demographic characteristics; age, gender and postcode. Further annual income and employment status were also uncorrelated with the total modified information overload score. This contrasts with Khaleel et al., who reported that low socioeconomic status is positively associated with information overload [12] based on studies they included in their scoping analysis. Overall, the scale could be applied to our population in the perioperative setting and yielded important information to guide information flow to parents.

Thematic analysis of the semi-structured interviews revealed that most parents did not feel overloaded with information. Some parents did report feeling overwhelmed when different staff presented differing or contradictory information, and this was identified as a barrier (conflicting information). This can be closely linked to another barrier around conflicting information between what is being given in the hospital and information that parents have accessed via the internet.

Another barrier was poor communication by staff. Parents gave examples of information, which was not presented at the parent's comprehension level, not being tailored to their child or being delivered too fast. This is important to note since we only interviewed parents with English as a first language, and given we found that information overload was significantly higher in those with a language other than English at home, this barrier would have an even greater impact on such families.

When questioned on suggestions to help facilitate communication, most parents in the interviews said it was helpful to have written information to take home; this was also reflected in the follow-up survey responses where 77% indicated a preference for written information, 9% verbal and 14% both verbal and written. Another suggestion from the parent interviews was to have take-home care information which was less generic and more personalised.

This study assessed a modified Information Overload scale for use in parents/carers of paediatric patients undergoing any elective surgical procedures and explored information overload among parents/carers through qualitative semi-structured interviews. A scoping review of health information overload among health consumers [12] reported on 22 studies, of which 10 focused on cancer information overload (CIO) with nothing in the perioperative space even though this may be of particular importance given parents/patients are given a lot of information at a stressful time, with the added risks of drug misuse and inadequate postoperative care.

## Limitations

The study was conducted at a single centre. However, our institution is the only paediatric tertiary referral centre in Western Australia, and it serves a wide range of patients from both metropolitan and regional areas. The interviewees were English-speaking parents only due lack of funding for interpretation services for research. It is a limitation of the study that information on health literacy was not collected in the survey and that purposive sampling of those with

high overload was not used for the qualitative study sampling. This work did not include patients having emergency surgery where time to receive and digest complex information is shorter and often received only verbally and at time when an individual's information-overload threshold is likely to be lowered [27].

Our results show that parents of elective surgical parents are not suffering from information overload in general, but they do want more information on immediate and late postoperative recovery. This study also highlights the need to take extra care when communicating with families of vulnerable population groups, including those who are non-native English speakers or who may not be university educated. By doing so, staff can make a meaningful difference to the patient experience and their understanding of the information presented to them. Future work in the elective surgery setting should focus on culturally and linguistically diverse patient groups and the level of information overload for this population, which would necessitate alternate communication strategies. The information overload scale may be useful in such research to identify those with the highest levels of overload. Studying the information overload in parents of children undergoing emergency surgery would also be worthwhile.

## Supporting information

**S1 File. POLARBEAR interview guide.**
(PDF)

**S2 File. Exploratory factor analysis.**
(PDF)

**S3 File. Additional results figures and tables.**
(PDF)

**S4 File. Participant questionnaire.**
(PDF)

## Acknowledgments

The authors acknowledge Ms Sharolin Boban for her contribution to conducting the semi-structured interviews.

## Author Contributions

**Conceptualization:** Aine Sommerfield, David Sommerfield, Kenneth Lee, R. Nazim Khan, Britta S. von Ungern-Sternberg.

**Data curation:** Daisy Evans, Megan Dodd, Emily Bell, Simone Gonsalves, R. Nazim Khan.

**Formal analysis:** Daisy Evans, Megan Dodd, Emily Bell, Simone Gonsalves, R. Nazim Khan.

**Funding acquisition:** Aine Sommerfield, Britta S. von Ungern-Sternberg.

**Investigation:** Megan Dodd, Emily Bell, Simone Gonsalves.

**Methodology:** Aine Sommerfield, David Sommerfield, Kenneth Lee, Daisy Evans, R. Nazim Khan, Britta S. von Ungern-Sternberg.

**Project administration:** Aine Sommerfield, David Sommerfield, R. Nazim Khan, Britta S. von Ungern-Sternberg.

**Resources:** David Sommerfield, Britta S. von Ungern-Sternberg.

**Supervision:** Aine Sommerfield, David Sommerfield, Kenneth Lee, R. Nazim Khan, Britta S. von Ungern-Sternberg.

**Visualization:** Daisy Evans.

**Writing – original draft:** Aine Sommerfield, David Sommerfield, Daisy Evans, R. Nazim Khan, Britta S. von Ungern-Sternberg.

**Writing – review & editing:** Aine Sommerfield, David Sommerfield, Kenneth Lee, Daisy Evans, Megan Dodd, Emily Bell, Simone Gonsalves, R. Nazim Khan, Britta S. von Ungern-Sternberg.

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
