## [Decision Letter · Decision Letter 0]

8 Jan 2024

PONE-D-23-36082Information overload and parental perspectives on information provided to parents/carers of paediatric patients undergoing elective surgical procedures.PLOS ONE

Dear Dr. Sommerfield,

Thank you for submitting your manuscript to PLOS ONE. After careful consideration, we feel that it has merit but does not fully meet PLOS ONE’s publication criteria as it currently stands. Therefore, we invite you to submit a revised version of the manuscript that addresses the points raised during the review process.

We look forward to receiving your revised manuscript.

Kind regards,

Boyen Huang, DDS, MHA, PhD

Academic Editor

PLOS ONE

Reviewers' comments:

Reviewer's Responses to Questions

**Comments to the Author**

1. Is the manuscript technically sound, and do the data support the conclusions?

Reviewer #1: Partly

Reviewer #2: Yes

2. Has the statistical analysis been performed appropriately and rigorously? 

Reviewer #1: No

Reviewer #2: N/A

3. Have the authors made all data underlying the findings in their manuscript fully available?

Reviewer #1: Yes

Reviewer #2: Yes

4. Is the manuscript presented in an intelligible fashion and written in standard English?

Reviewer #1: No

Reviewer #2: Yes

5. Review Comments to the Author

Reviewer #1: Information overload and parental perspectives on information provided to parents/carers of paedicatric patients undergoing elective surgical procedures.

First author: Sommerfield, A.

Corresponding author: von Ungern-Sternberg, BS.

Country: Australia

PLOS ONE Review questions:

1. Is the manuscript technically sound, and do the data support the conclusions?

In part, reasons are elaborated on in 5. Review comments to the authors.

2. Has the statistical analysis been performed appropriately and rigorously?

No, possibly analyses are appropriate, but unclear reporting in methods and results make it difficult to ascertain. Reasons are further elaborated on under 5. Review comments to the authors.

3. Have the authors made all data underlying the findings in their manuscript fully available?

Yes

4. Is the manuscript presented in an intelligible fashion and written in standard English?

No. However, the authors are clearly proficient in English, so “No” is related to the reporting of statistical analyses and results. The manuscript would profit from a careful reading by a statistician or others with similar competencies. The STROBE reporting guidelines may also be helpful in this respect.

5. Review comments to the Authors

The paper investigates information overload as reported by parents or carers of pediatric patients undergoing elective surgical procedures. A strength of the study is that both quantitative (survey and interview) and qualitative (interview) data are collected. The study may be perceived as a mixed methods (MM) study. If it was planned as such, and depending on the chosen MM design, all parts of the paper will need major revision. If not, there are still other issues that warrant major revisions. These are addressed below.

Introduction

• The authors may consider using a more comprehensive definition of health literacy, possibly in line with Sørensen et al. (2012), DOI: 10.1186/1471-2458-12-80.

• The research gap should be stated clearer. It is not explicitly communicated whether research exist that would answer the research question.

• Some statements in the last paragraph miss references. This concerns the sentences starting with “Parents attending …..”, and “Furthermore, the perioperative time ….” .

• The introduction should give the rationale for collecting both quantitative and qualitative data. Are the two approaches meant to give complementary data, or are there other reasons? E.g. on page 5, the survey is denoted as “stage 1” indicating a sequential design.

• STROBE recommend that objectives are stated as part of the introduction.

Materials and methods

Objectives

• I cannot see that the paper research and reports on the utility of the scale (objective 1).

• Although it may be considered as a separate research question, objective 2 could be expanded to: …identify the level and explore factors associated with …

Survey; material and analyses

• Information should be given as to when and where parents/carers were approached, e.g. where the poster was located.

• Any categorizations of variables (like for post code) should be described.

• If a statistical power analysis was performed before study start the result should be reported.

• How the regression model-building was performed is not clear. 1) Regarding the inclusion of interaction terms, the phrase “…as appropriate based on data exploration.”, needs to be elaborated. 2) It is unclear whether interaction terms were included or not in the stepwise regression analysis.

• Note that “removing stepwise” is not commonly used to describe a backward stepwise regression.

• In the discussion (lines 324-327) the authors write that low education level, low health literacy, poor searching skills, greater concern about information quality, low socioeconomic status, and race (Hispanic and non-Hispanic black) are positively associated with information overload with reference to Khaleel et al. (2020). In methods (page 8) only education is highlighted. Why?

Interview; material and analyses

• Information should be given on when the interviews were performed (time since the operation).

• The authors may consider providing the interview guide as an additional file.

• Lines 150 (The demographic …) – 151: It is unclear whether demographic data was collected during the interview or was retrieved from existing survey data.

• Lines 152 (Thematic ….) - 153 is a description of analyses and should be moved to the analysis section (or removed if already given).

• It should be stated explicitly what phenomena barriers and facilitators points to, either in methods or results – whatever is the more appropriate.

Results

Survey

• Lines 191 -194 (until A total of…) describes methods and should be moved or deleted.

• Is it important that it is 15 %? Does the number indicate a rule for inclusion of reported data?

• Mean and standard deviation of the sum score should be provided.

• The phrase “insignificant covariates” should be used with caution. “Not statistically significant” would be more appropriate.

• Results from simple (unadjusted) regression analyses should be considered presented in an additional file. This will increase the transparency of the analyses, in particular as a stepwise regression procedure was used.

• It is unclear whether all results in table 4 comes from a regression model including interaction terms between aboriginal status and education. If that is the case, the interpretation of the estimated main effects is incorrect.

Interview

• Use of the reporting guideline (COREQ) should be stated under methods.

• Lines 253 (Willing parents…) - 254 may be considered moved to methods.

• A detail, but both the terms facilitators and enablers are used.

• If barriers, facilitators and overloaded were the main themes identified this should be stated explicitly. It is now not clear whether these were identified or pre-defined.

• Subheadings (barriers etc.) may increase the readability of this section.

Discussion

• The first paragraph should give a much shorter and less detailed overview of the results

• Subheadings may increase readability.

• Strengths: Suggest more explicit statements of what was gained by retrieving both quantitative and qualitative data.

• It is a limitation of the study that information on health literacy was not collected in the survey.

Reviewer #2: Thank you for the opportunity to review the important work on understanding children and their family caregivers experience with the health education providers delivered. The triangulation of the data and analysis were strengths of the study. I have two minor questions

1. The transformation of the CIO measurement by simply removing the cancer word is not usually recommended. Authors should report the validity and the reliability of the new scale and whether the new scale has been pilot tested.

2. It would be helpful to know what types of surgeries the children are undergoing and other demographic characteristics and their primary diagnosis when interpreting the results. The null findings is likely due to the non-cancer related diagnosis.

6. PLOS authors have the option to publish the peer review history of their article (what does this mean?). If published, this will include your full peer review and any attached files.

Reviewer #1: No

Reviewer #2: No

---

## [Author Response · Author response to Decision Letter 0]

7 Feb 2024

Below are our responses to the points raised by the reviewers. 

REVIEWER #1: 

Is the manuscript presented in an intelligible fashion and written in standard English?

No. However, the authors are clearly proficient in English, so “No” is related to the reporting of statistical analyses and results. The manuscript would profit from a careful reading by a statistician or others with similar competencies. The STROBE reporting guidelines may also be helpful in this respect.

Authors’ response: One of the authors is an experienced statistician Dr Nazim Khan who has clarified sections in the revised manuscript.

Review comments to the Authors

The paper investigates information overload as reported by parents or carers of pediatric patients undergoing elective surgical procedures. A strength of the study is that both quantitative (survey and interview) and qualitative (interview) data are collected. The study may be perceived as a mixed methods (MM) study. If it was planned as such, and depending on the chosen MM design, all parts of the paper will need major revision. If not, there are still other issues that warrant major revisions. These are addressed below.

Authors’ response: We have addressed the issues raised in the revised manuscript.

Introduction

The authors may consider using a more comprehensive definition of health literacy, possibly in line with Sørensen et al. (2012), DOI: 10.1186/1471-2458-12-80.

Authors’ response: The definition has been updated in the revision. 

The research gap should be stated clearer. It is not explicitly communicated whether research exist that would answer the research question.

Authors’ response: This has now been explicitly stated.

Some statements in the last paragraph miss references. This concerns the sentences starting with “Parents attending …..”, and “Furthermore, the perioperative time ….” .

Authors’ response: References have been added.

The introduction should give the rationale for collecting both quantitative and qualitative data. Are the two approaches meant to give complementary data, or are there other reasons? E.g. on page 5, the survey is denoted as “stage 1” indicating a sequential design.

Authors’ response: This has now been clarified in the revised manuscript. The quantitative and qualitative information is meant to give complementary data. First collecting overarching data and then diving deeper in the issues.

STROBE recommend that objectives are stated as part of the introduction.

Authors’ response: This has now been clarified in the introduction.

Materials and methods

Objectives

I cannot see that the paper research and reports on the utility of the scale (objective 1).

Authors’ response: This had now been further clarified in the revised manuscript.

Although it may be considered as a separate research question, objective 2 could be expanded to: …identify the level and explore factors associated with …

Authors’ response: This is an interesting idea that could be pursued in future research. 

Survey; material and analyses

Information should be given as to when and where parents/carers were approached, e.g. where the poster was located.

Authors’ response: This has now been clarified in the revised manuscript.

Any categorizations of variables (like for post code) should be described.

Authors’ response: Thank you for pointing out this omission, we have added further detail as to the classification of postcodes into metropolitan (metro) or rural status to our methods section as follows: “Postcodes were classified as metropolitan (metro), regional, or remote based on the Australian Bureau of Statistics Statistical Geography Standard Correspondences Postcode 2017 to Remoteness Area 2016. Postcodes classified as being in “Major Cities of Australia” were grouped under “metro”, postcodes classified as “Inner Regional Australia” or “Outer Regional Australia” were grouped under “regional”, and those classified as “Remote Australia” or “Very Remote Australia” were grouped under “remote”.”

If a statistical power analysis was performed before study start the result should be reported.

Authors’ response: Given that there is no previous data to base our calculations on, we were unable to perform a power calculation. Further, as a key objective of the quantitative component of our study was to estimate prevalence of information overload, we used conservative parameters to estimation sample size based on population proportion. We have updated our manuscript to clarify this.

How the regression model-building was performed is not clear. 1) Regarding the inclusion of interaction terms, the phrase “…as appropriate based on data exploration.”, needs to be elaborated. 2) It is unclear whether interaction terms were included or not in the stepwise regression analysis. Note that “removing stepwise” is not commonly used to describe a backward stepwise regression.

Authors’ response: The Data Analysis methods have been revised for clarity. Additionally, relevant interaction plots have been included as supplementary figures. The model-building methodology now reads: “Variables considered in the regression analysis included participants’ demographic characteristics (gender, age group, Aboriginal heritage, whether a language other than English is spoken at home, education level, employment status, income, postcode remoteness) and whether or not the respondent or any of their children had previously had a similar surgery. Interaction terms between variables were included based on exploration of interaction plots and cross-tables. Backwards stepwise regression was used to reduce the model to significant covariates, where statistical significance was taken at p<0.05.”

In the discussion (lines 324-327) the authors write that low education level, low health literacy, poor searching skills, greater concern about information quality, low socioeconomic status, and race (Hispanic and non-Hispanic black) are positively associated with information overload with reference to Khaleel et al. (2020). In methods (page 8) only education is highlighted. Why?

Authors’ response: We did state that the interaction between Aboriginal status and education is significant, and this relates to the finding by Khaleel et al. regarding race. However, in our data we did not find a correlation with annual household income and employment status, which are proxies for socioeconomic status. We have made this comparison with Khaleel et al. more explicit in the discussion.

Information should be given on when the interviews were performed (time since the operation).

Authors’ response: The interviews were all conducted between 20 and 59 days following the operation. A statement on this has been added to the manuscript. 

The authors may consider providing the interview guide as an additional file.

Authors’ response: The ethics committee approved Interview Guide has been included as suggested. 

Lines 150 (The demographic …) – 151: It is unclear whether demographic data was collected during the interview or was retrieved from existing survey data.

Authors’ response: The demographic data was collected during the interview. A sentence has been added to clarify this. 

Lines 152 (Thematic ….) - 153 is a description of analyses and should be moved to the analysis section (or removed if already given).

Authors’ response: This has been moved to analysis. 

It should be stated explicitly what phenomena barriers and facilitators points to, either in methods or results – whatever is the more appropriate.

Authors’ response: The significance is addressed in the discussion section.

Lines 191 -194 (until A total of…) describes methods and should be moved or deleted.

Authors’ response: Thank you, these have been removed as they are a repeat of what is already in the methods.

Is it important that it is 15 %? Does the number indicate a rule for inclusion of reported data?

Authors’ response: It is not important that is less than 15% particularly, we have re-phrased this sentence for clarity. It now reads: “Of these, 2 participants did not respond to any of the survey questions following consent, and were subsequently excluded from analysis…”

Mean and standard deviation of the sum score should be provided.

Authors’ response: For consistency, the median (LQ, UQ) of the sum score has been reported as well as the mean and standard deviation. The median (LQ, UQ) of the total score has been added to Table 3, and the mean, SD, minimum and maximum scores reported in the text.

The phrase “insignificant covariates” should be used with caution. “Not statistically significant” would be more appropriate.

Authors’ response: The relevant sentence has been amended in the text, and now reads: “Following backwards stepwise regression, the estimated coefficients…”

Results from simple (unadjusted) regression analyses should be considered presented in an additional file. This will increase the transparency of the analyses, in particular as a stepwise regression procedure was used.

Authors’ response: We assume the reviewer is referring to the full model before covariate reduction, given that single-variable regression models are generally misleading since they don’t consider interactions and effects of covariates. To address this, the table of estimates from the model before covariate reduction has been included as an appendix to the manuscript.

It is unclear whether all results in table 4 comes from a regression model including interaction terms between aboriginal status and education. If that is the case, the interpretation of the estimated main effects is incorrect.

Authors’ response: The manuscript text under Results – Regression analysis has been amended for clarity based on this feedback. The discussion has also been appropriately amended.

Use of the reporting guideline (COREQ) should be stated under methods.

Authors’ response: A statement on the use of COREQ has been added to the methods section. 

Lines 253 (Willing parents…) - 254 may be considered moved to methods.

Authors’ response: this sentence moved to methods section as suggested. 

A detail, but both the terms facilitators and enablers are used.

Authors’ response: We have changed faciltators in the table to enablers which to ensure that only the term enabler is used throughout the manuscript

If barriers, facilitators and overloaded were the main themes identified this should be stated explicitly. It is now not clear whether these were identified or pre-defined.

Authors’ response: These were identified and this has been clarified in the revised manuscript.

Subheadings (barriers etc.) may increase the readability of this section.

Authors’ response: We respectfully disagree since it would take away the flow of the discussion.

The first paragraph should give a much shorter and less detailed overview of the results

Authors’ response: This section has been revised.

Strengths: Suggest more explicit statements of what was gained by retrieving both quantitative and qualitative data.

Authors’ response: This has been clarified in the revised manuscript.

It is a limitation of the study that information on health literacy was not collected in the survey.

Authors’ response: A statement to this affect has been added to the Limitation section at the end of the manuscript.

Reviewer #2: 

The transformation of the CIO measurement by simply removing the cancer word is not usually recommended. Authors should report the validity and the reliability of the new scale and whether the new scale has been pilot tested.

Authors’ response: We followed the methodology used in the article of Obamiro K, Lee K. (Information overload in patients with atrial fibrillation: Can the cancer information overload (CIO) scale be used? Patient Educ Couns. 2019;102(3):550-4).We never intended to validate the scale. 

It would be helpful to know what types of surgeries the children are undergoing and other demographic characteristics and their primary diagnosis when interpreting the results. The null findings is likely due to the non-cancer related diagnosis.

Authors’ response: We agree that it would have been ideal to consider further characteristics and surgery type in the analysis, however we did not collect this information from participants. All participant characteristic information that was collected has been included in the analysis. The word cancer, in line with the article from Obamiro K, Lee K. (2019), was removed from the scale and replaced with caring for your child after surgery. Therefore, the type of surgery was not collected as we were looking for an overall impression of parents experience regardless of surgery type and diagnosis.

---

## [Decision Letter · Decision Letter 1]

27 Feb 2024

PONE-D-23-36082R1Information overload and parental perspectives on information provided to parents/carers of paediatric patients undergoing elective surgical procedures.PLOS ONE

Dear Dr. Sommerfield,

Thank you for submitting your manuscript to PLOS ONE. After careful consideration, we feel that it has merit but does not fully meet PLOS ONE’s publication criteria as it currently stands. Therefore, we invite you to submit a revised version of the manuscript that addresses the points raised during the review process.

In addition to responding to Reviewer 1's comments on the presentation of the references, guidelines and tables, please thoroughly address the statistical analysis concerns raised by both Reviewers 1 and 2.

We look forward to receiving your revised manuscript.

Kind regards,

Boyen Huang, DDS, MHA, PhD

Academic Editor

PLOS ONE

Reviewers' comments:

Reviewer's Responses to Questions

**Comments to the Author**

1. If the authors have adequately addressed your comments raised in a previous round of review and you feel that this manuscript is now acceptable for publication, you may indicate that here to bypass the “Comments to the Author” section, enter your conflict of interest statement in the “Confidential to Editor” section, and submit your "Accept" recommendation.

Reviewer #1: (No Response)

Reviewer #2: (No Response)

2. Is the manuscript technically sound, and do the data support the conclusions?

Reviewer #1: Yes

Reviewer #2: No

3. Has the statistical analysis been performed appropriately and rigorously? 

Reviewer #1: Yes

Reviewer #2: No

4. Have the authors made all data underlying the findings in their manuscript fully available?

Reviewer #1: Yes

Reviewer #2: No

5. Is the manuscript presented in an intelligible fashion and written in standard English?

Reviewer #1: Yes

Reviewer #2: Yes

6. Review Comments to the Author

Reviewer #1: Thanks for the thorough review of my previous comments.

Only a few concerns remain.

These are:

A reference should be provided to the tool used for the statistical power sample calculations.

The COREQ guideline was used for the reporting of the qualitative data. Likewise, the reporting of the quantitative data should be checked against the STROBE guideline for observational studies (https://www.ncbi.nlm.nih.gov/pmc/articles/PMC2020496/). If this is already the case, it should be stated in the manuscript.

Sorry for not being clear but reporting of unadjusted estimates may be called for. See comments to item 16 in the STROBE guideline.

Heading table 4: “from a multiple” is repeated.

If the regression model reported in table 4 included both main effects and interaction effects, the main effects for e.g. Education should be interpreted as effects at the reference level of Aboriginal status. My apologies if I have misunderstood, but this may not be properly reflected in the text.

Comment on a data driven variable selection vs using background information:

While using background information is often advocated it is also recognized that this may be difficult if there is no supporting strong theory. The exploratory approach argued by the authors is therefore an acceptable solution (https://www.ncbi.nlm.nih.gov/pmc/articles/PMC5969114/).

Reviewer #2: The key measurement in the study lacked psychomatric validity, and the statistical analysis was lacking rigor, as it is unknown whether the proposed advanced statistical analysis regression met its statistical assumptions before interpreting the findings.

7. PLOS authors have the option to publish the peer review history of their article (what does this mean?). If published, this will include your full peer review and any attached files.

Reviewer #1: No

Reviewer #2: No

---

## [Author Response · Author response to Decision Letter 1]

17 Mar 2024

REVIEWER #1:

A reference should be provided to the tool used for the statistical power sample calculations.

Authors’ Response: The sample size estimator used can be found at . We have added this reference into the manuscript.

The COREQ guideline was used for the reporting of the qualitative data. Likewise, the reporting of the quantitative data should be checked against the STROBE guideline for observational studies (https://www.ncbi.nlm.nih.gov/pmc/articles/PMC2020496/). If this is already the case, it should be stated in the manuscript.

Authors’ Response: The reporting has been checked against the STROBE guidelines. The following sentence has been added to the results section: “STROBE guidelines were considered in the reporting of results” and the STROBE guidelines have been referenced.

Sorry for not being clear but reporting of unadjusted estimates may be called for. See comments to item 16 in the STROBE guideline.

Authors’ Response: In statistical modelling, the unadjusted estimates can be misleading and in fact be entirely incorrect. Adjusted estimates allow us to isolate the effect of the key variable(s) of interest after adjusting for effects of confounding variables. Nonetheless, plots of raw unadjusted mean and bootstrapped 95% CI which were used in investigation of interaction terms are included in the supplement.

Heading table 4: “from a multiple” is repeated.

Authors’ Response: Thank you for pointing this out. We have corrected it.

If the regression model reported in table 4 included both main effects and interaction effects, the main effects for e.g. Education should be interpreted as effects at the reference level of Aboriginal status. My apologies if I have misunderstood, but this may not be properly reflected in the text.

Authors’ Response: To facilitate the interpretation of this interaction term, we have amended the figure shown in the main text (Fig1, reproduced below). 

When interaction terms are used in statistical modelling, care must be taken not to interpret the main effects of each variable in isolation. Plots of the estimated effects can assist in interpretation. From the figure, focussing on the non-Aboriginal or Torres Strait Islander participants (represented by red circles), we see a general trend towards lower modified 5-item CIO score with increasing education, with a slight uptick for those with postgraduate university degrees compared to undergraduate university degrees. As you correctly interpreted, the main effects of education can be interpreted as the effect of education level on the reference level of Aboriginal or Torres Strait Islander (which is “No”). When considering the interaction terms, the level of the main effects of Aboriginal or Torres Strait Islander, and of the main effects of Education, and of the interaction term must be considered. Considering the figure can assist with this interpretation – from Table 4 (also reproduced below), the estimate of the interaction term [Aboriginal or Torres Strait Islander = Yes] * [Education = High school completion] is statistically significantly different from 0 (Est 3.4 95% CI (0.55, 6.26)). From the figure, we see that this can be interpreted as: for those participants with [Education = High school completion], those with Aboriginal or Torres Strait Islander heritage have significantly higher Modified 5-item CIO Score. However, for those participants with [Education= Did not complete high school] (that is, the reference level of Education), Aboriginal and Torres Strait Islander participants do not have a significantly different Score to non-Aboriginal or Torres Strait Islander participants (Est -0.05 95% CI (-1.92, 1.82)).

Variable Estimate Std. Error p-value 95% C.I.

 2.5% 97.5%

Intercept 9.94 0.44 <0.001 9.08 10.8

Aboriginal or Torres Strait Islander

 No REF 

 Yes -0.05 0.95 0.958 -1.92 1.82

 Prefer not to say 0.21 2.4 0.931 -4.5 4.92

Education Did not complete high school REF 

 High school completion -0.48 0.53 0.362 -1.53 0.56

 Technical/TAFE qualification -1.21 0.48 0.013 -2.16 -0.26

 Undergraduate University degree -1.63 0.51 0.002 -2.64 -0.62

 Postgraduate university degree -1.12 0.52 0.031 -2.15 -0.1

LOTE No REF 

 Yes 0.98 0.28 <0.001 0.42 1.53

Aboriginal status * Education Yes * High school completion 3.4 1.45 0.020 0.55 6.26

 Yes * Technical/TAFE qualification 0.67 1.37 0.624 -2.02 3.36

 Prefer not to say * Technical/TAFE qualification -0.93 3.38 0.782 -7.58 5.71

To further assist with interpretation, the interaction plot S1 Fig1 in the supplement (showing raw mean and bootstrapped 95% CI of the data, not model estimates) has also been adjusted to show points grouped by Education level on the x-axis and by Aboriginal or Torres Strait Islander heritage by shape/colour of the points. This plot of the raw means justifies the inclusion of the interaction term in the model at all.

Comment on a data driven variable selection vs using background information:

While using background information is often advocated it is also recognized that this may be difficult if there is no supporting strong theory. The exploratory approach argued by the authors is therefore an acceptable solution (https://www.ncbi.nlm.nih.gov/pmc/articles/PMC5969114/).

Authors’ Response: We thank the reviewer for confirming this.

REVIEWER #2:

The key measurement in the study lacked psychometric validity, and the statistical analysis was lacking rigor, as it is unknown whether the proposed advanced statistical analysis regression met its statistical assumptions before interpreting the findings.

Authors’ Response: 

Psychometric validity

The original CIO scale was validated , although we recognise that we have used an altered version of the scale in a different test population. There is precedent for this alteration of the 5-item CIO scale (see Obamiro and Lee 2019 [doi.org/10.1016/j.pec.2018.10.005], Breyton et al 2023 [

Doi.org/10/1016/j.pec.2023.107672]). To address the fact that the validity of the original CIO scale does not necessarily translate to the validity of our modified scale, we performed tests of reliability and validity (based on the methods used in Obamiro and Lee 2019), including calculation of Cronbach’s alpha to assess internal consistency and reliability, and exploratory factor analysis (EFA) to assess the factor structure of our modified 5-item CIO score for caring for children after surgery. 

A further aspect of construct validity, differentiation between known groups (Boateng et al. Best Practices for Developing and Validating Scales for Health, Social, and Behavioral Research: A Primer.

Front. Public Health 2018 6:149.doi: 10.3389/fpubh.2018.00149), can be demonstrated by our regression analysis of the modified 5-item CIO total score against participant characteristic details. The fact that the modified 5-item CIO score is significantly associated with participants’ education level (decreasing with higher education), and increases for participants who speak primarily a language other than English, confirms expected associations between the score and characteristics theoretically associated with information overload.

The assessment of the psychometric properties of the modified 5-item CIO used here has now been explicitly stated in the methods and results, and details have been included in an additional supplementary document. 

We recognise that the methods considered here are not the only requirements to establish validity of a scale, but we believe that the findings of our study on information overload in parents of children undergoing surgery are relevant and important all the same, and hope that our expansion on the properties of the modified 5-item CIO scale is satisfactory to address these comments.

Regression assumptions

We have now included a statement on the model assumptions in the methods ("Model assumptions were verified by inspection of diagnostic plots.”) and the results sections (“Model diagnostics confirmed that the normal linear model assumptions were verified.”). The diagnostics plots are included below for information. The residuals are homogeneous and do not display evidence of nonlinearity (from the plot of residuals vs fitted values), and are approximately normally distributed (from visual inspection of the QQ plot of residuals).

---

## [Decision Letter · Decision Letter 2]

30 Apr 2024

PONE-D-23-36082R2Information overload and parental perspectives on information provided to parents/carers of paediatric patients undergoing elective surgical procedures.PLOS ONE

Dear Dr. Sommerfield,

Thank you for submitting your manuscript to PLOS ONE. After careful consideration, we feel that it has merit but does not fully meet PLOS ONE’s publication criteria as it currently stands. Therefore, we invite you to submit a revised version of the manuscript that addresses the points raised during the review process.

We look forward to receiving your revised manuscript.

Kind regards,

Boyen Huang, DDS, MHA, PhD

Academic Editor

PLOS ONE

Reviewers' comments:

Reviewer's Responses to Questions

**Comments to the Author**

1. If the authors have adequately addressed your comments raised in a previous round of review and you feel that this manuscript is now acceptable for publication, you may indicate that here to bypass the “Comments to the Author” section, enter your conflict of interest statement in the “Confidential to Editor” section, and submit your "Accept" recommendation.

Reviewer #1: (No Response)

Reviewer #2: All comments have been addressed

Reviewer #3: (No Response)

2. Is the manuscript technically sound, and do the data support the conclusions?

Reviewer #1: Yes

Reviewer #2: Yes

Reviewer #3: No

3. Has the statistical analysis been performed appropriately and rigorously? 

Reviewer #1: No

Reviewer #2: Yes

Reviewer #3: I Don't Know

4. Have the authors made all data underlying the findings in their manuscript fully available?

Reviewer #1: Yes

Reviewer #2: No

Reviewer #3: No

5. Is the manuscript presented in an intelligible fashion and written in standard English?

Reviewer #1: Yes

Reviewer #2: No

Reviewer #3: No

6. Review Comments to the Author

Reviewer #1: Re-re-review: Information overload and parental perspectives on information provided to parents/carers of paedicatric patients undergoing elective surgical procedures.

Thanks for your thorough consideration of my comments.

Two comments:

I agree that unadjusted estimates may be misleading due to observed and unobserved confounders (and of course other modelling issues). However, the reason for providing both are as is pointed out in the STROBE guideline, 16a: “Readers can compare unadjusted measures of association with those adjusted for potential confounders and judge by how much, and in what direction, they changed.” S2 table 1 provides estimates from the full model, including interaction terms, before the backward stepwise procedure, and so does not give unadjusted (crude) estimates. My conclusion was therefore that the statistical analysis had not been performed appropriately and rigorously. This may however be an issue best decided by the editor.

The authors write in their response:” When interaction terms are used in statistical modelling, care must be taken not to interpret the main effects of each variable in isolation”. This was exactly my point, so thanks for the clarification.

Reviewer #2: Thank you for your important work on addressing parents experiences particularly information overload in healthcare settings.

Reviewer #3: I have been invited to review this interesting paper which has previously been reviewed twice by others. I agree that Information Overload is a potentially important issue for parents caring for children undergoing surgery.

I have some concerns about the data collected, the reporting of the methods used for data collection and analyses, the reporting of the analyses (with a focus on the qualitative analysis) and the interpretation of the results which has led me to answer “No” to most of the questions. I answered Don't Know to the Statistical Analysis question as I have not been able to allocate sufficient time to review the statistical analysis in sufficient depth to offer a fair evaluation. Below is a summary of my concerns and suggestions from revisions.

Adaption of the Cancer Information Overload measure

The wording in your adapted measure appears to be significantly different from the original version, beyond a simple switch of the word “cancer” for “caring for your children after surgery” as stated in line 141. This leads me to have concerns about whether the modified scale is measuring information overload as intended or could potentially be measuring something else. There doesn’t appear to be a report of any pre-piloting to provide confirmation that the items are understand as intended by parents.

For example:

Item 2 (original) “There is so much cancer information, I don't even care to hear new things about cancer”.

Item 2 (modified) “It has gotten to the point where I don’t even care to hear new information about caring for my child after surgery”.

It would also be helpful to have a statement about the readability of the modified items to ensure that the items are comprehensible to parents of varying educational attainment levels. It would be possible to test readability retrospectively.

The EFA was used to explore the factor structure of the scale. It would be helpful to have a sentence in the methods or results that links the EFA to the decision to sum the 5 items to form a one factor scale.

Qualitative study

More information is needed to enable the reader to evaluate the qualitative study data. It would be helpful to have a brief reference to the authors’ epistemological stance along with a statement of reflexivity which is helpful to understand the relationship of the researchers to the data collection and insight generation process.

More information is needed about the sampling strategy. 35.6% (n=136) agreed to be interviewed, it would be good to know what the strategy was to decide who to contact – was a purposive or consecutive sampling approach taken and with what justification. It would be great to have a more detailed description of the analytical process undertaken to develop the coding framework – including reference to whether those who completed the analysis were the same or different people to those who conducted the interviews, was coding inductive or deductive, how did you arrive at the key themes of ‘barriers’ and ‘facilitators’ and what did the barriers and facilitators relate to – information overload or general communication or something else?

Presentation of quotes from participants should ideally be at a minimum labelled with a participant ID so that the reader can be reassured that the quote come from a number of participants. It can also be useful to have other descriptives e.g. gender, or education level so that there is some contextual information to help interpret the quotes offered.

Relationship between the quantitative and qualitative study

The two studies feel unconnected at the moment, and I wonder if more could be done to connect the findings of the two studies. The qualitative study appears to be a more general description of communication experiences of parents rather than a more in-depth exploration of the experience of information overload. This may reflect the low prevalence of information sample in the questionnaire study from which the qualitative sample was drawn. Purposive sampling to specifically invite questionnaire participants with higher CIO scores to the interview study may have enabled this in-depth exploration and I think this is a potentially limitation of the paper as it is presented.

Below are a few additional points that I hope you might find helpful. I found that in places the manuscript was not logically organised (e.g. description of all the variables used in the quant study).

Introduction

• Acknowledging the comment that there is little specific research that has been done to measure information burden in parents whose child has under gone elective surgery, I wonder if there may still be benefit to adding more contextual research that could give the reader some ideas about what factors might be influential here. E.g. Potential parental demographic, psychological variables, child medical details, HCP related factors. They are hinted at but not referenced e.g. line 99 and 100 or referenced in the discussion, line 397 Khaleel reference about association of SES with information overload.

Methods

• Line 180 recommend use of “variables” rather than “factors” as it is confusing to refer to factors straight after mentioning EFA.

• It would add to clarity if there was a specific participants/ procedure section and separate measures section. Currently reference to some of the measures only appear in the analyses/ results e.g. prior experience of health care and language spoken at home – with the abbreviation LOTE referenced without introduction as far as I can see. A copy of the questionnaire could be added as a supplementary document for additional information

Results:

• Regression analysis – description feels out of order by starting with reference to the interaction results – clarity / logical order would be improved if started with reference to the unadjusted results of all the variables (table I think is in supplementary) and go from there, building up to the interaction results. I haven't unfortunately had time to consider the statistical analysis in detail so I have answered "Don't know" to the question related that asks me to assess whether the statistical analysis has been completed appropriately and rigorously.

• Title of table 6 could be revised for clarity – I would remove reference to stage 3 and add something that tells the reader explicitly what the barriers and enablers specifically relate to e.g. general communication or avoidance of information overload?

• It would be great to see themes evidenced with quotes alongside in the table

• I think some of the sub codes could be revised for greater conceptual clarity:

e.g. Own Research or Experience - seems to relate to two different things: 1) conflict/ confusion arising from parents finding information from the internet which is different from what they have been told by hospital staff and 2) problems related to individual characteristics of parents e.g. ADHD.

e.g. Barriers: poor communication- appears to give a description of HCP communication that was perceived to be ineffective whereas when it comes to Enablers: “good” HCP communication is sub-divided to into different categories e.g. written, verbal, rapport.

Discussion

In this section it would be great to see a more detailed interpretation/ assessment of the validity, significance, and implications of the findings. For example, do the authors think that the adapted CIO could be improved upon, could/ should be used going forward? Was the low level of information overload in the sample an expected or unexpected finding? What future work could follow these studies? For example, what about the experiences of parents whose children have emergency surgery?

7. PLOS authors have the option to publish the peer review history of their article (what does this mean?). If published, this will include your full peer review and any attached files.

Reviewer #1: No

Reviewer #2: No

Reviewer #3: No

---

## [Author Response · Author response to Decision Letter 2]

30 May 2024

Below are our responses to the points raised by the reviewers. 

REVIEWER #1:

I agree that unadjusted estimates may be misleading due to observed and unobserved confounders (and of course other modelling issues). However, the reason for providing both are as is pointed out in the STROBE guideline, 16a: “Readers can compare unadjusted measures of association with those adjusted for potential confounders and judge by how much, and in what direction, they changed.” S2 table 1 provides estimates from the full model, including interaction terms, before the backward stepwise procedure, and so does not give unadjusted (crude) estimates. My conclusion was therefore that the statistical analysis had not been performed appropriately and rigorously. This may however be an issue best decided by the editor.

Authors’ Response: We have now added a table of the unadjusted single-variable regression estimates to Supplement 2.

The authors write in their response:” When interaction terms are used in statistical modelling, care must be taken not to interpret the main effects of each variable in isolation”. This was exactly my point, so thanks for the clarification.

Authors’ Response: We thank the reviewer for their comments on the clarity of the statistical modelling, which allowed us to improve the communication of the regression modelling results.

REVIEWER #2:

Thank you for your important work on addressing parents experiences particularly information overload in healthcare settings.

Authors’ Response: We thank the reviewer for their comments which led to the improvement of the quality of our manuscript.

REVIEWER #3:

I have been invited to review this interesting paper which has previously been reviewed twice by others. I agree that Information Overload is a potentially important issue for parents caring for children undergoing surgery.

I have some concerns about the data collected, the reporting of the methods used for data collection and analyses, the reporting of the analyses (with a focus on the qualitative analysis) and the interpretation of the results which has led me to answer “No” to most of the questions. I answered Don't Know to the Statistical Analysis question as I have not been able to allocate sufficient time to review the statistical analysis in sufficient depth to offer a fair evaluation. Below is a summary of my concerns and suggestions from revisions.

Authors’ Response: Thank you for your comments, our answers are presented one by one below.

Adaption of the Cancer Information Overload measure

The wording in your adapted measure appears to be significantly different from the original version, beyond a simple switch of the word “cancer” for “caring for your children after surgery” as stated in line 141. This leads me to have concerns about whether the modified scale is measuring information overload as intended or could potentially be measuring something else. There doesn’t appear to be a report of any pre-piloting to provide confirmation that the items are understand as intended by parents.

For example:

Item 2 (original) “There is so much cancer information, I don't even care to hear new things about cancer”.

Item 2 (modified) “It has gotten to the point where I don’t even care to hear new information about caring for my child after surgery”.

Authors’ Response: We apologise that we were unclear about the origins of the scale used in our study. It is based on the modified 5 item Cancer Information Overload Scale (Costa et al) and an 8-item atrial fibrillation information overload scale published by one of the authors on this manuscript (KL). The table below shows how these three scales compare: 

Atrial fibrillation scale (Obarimo and Lee 2019) Modified CIO scale (Costa et al 2015. Scale used by us

There are so many different recommendations about managing atrial fibrillation, it’s hard to know which ones to follow There are so many different recommendations about cancer, it’s hard to know which ones to follow. There are so many different recommendations about caring for your child after surgery, it’s hard to know which ones to follow

There is not enough time to do all of the things recommended to manage atrial fibrillation N/A

It has gotten to the point where I don’t even care to hear new information about atrial fibrillation There is so much cancer information, I don't even care to hear new things about cancer. It has gotten to the point where I don’t even care to hear new information about caring for my child after surgery

No one could actually do all of the atrial fibrillation management recommendations that are given N/A

Information about atrial fibrillation all starts to sound the same after a while There is so much information about cancer, it all starts to sound the same after a while. Information about caring for my child after surgery all starts to sound the same after a while

I forget most of the information about atrial fibrillation right after I hear it There is so much cancer information, I forget most cancer information right after I learn it. I forget most of the information about caring for my child after surgery right after I hear it

Most things I hear or read about atrial fibrillation seem pretty far-fetched N/A

I feel overloaded by the amount of information about atrial fibrillation I am supposed to know I feel overloaded by the amount of cancer information I am supposed to know. I feel overloaded by the amount of information about caring for my child after surgery I am supposed to know

We have made changes to the manuscript to make this clearer and called our scale an information overload scale throughout. Please note that we while we didn’t pre-audit the scale used, we did consult with our Anaesthesia Research Consumer Reference Panel which consists of a diverse group of parents/consumers, and they were happy with the scale. 

It would also be helpful to have a statement about the readability of the modified items to ensure that the items are comprehensible to parents of varying educational attainment levels. It would be possible to test readability retrospectively.

Authors’ Response: We work very closely with consumers in all our projects. Our consumer research reference panel is a panel with members of diverse backgrounds, including CALD members and indigenous members. All questions were designed with the help of our consumer partners to ensure readability for a wide range of parents. The 

The EFA was used to explore the factor structure of the scale. It would be helpful to have a sentence in the methods or results that links the EFA to the decision to sum the 5 items to form a one factor scale.

Authors’ Response: The score was summed to align with the original information overload score and previously published adjusted CIO scores. A sentence has been added to the results to reflect this:

Exploration of the factor structure indicated a one- or two-factor solution; however, the individual modified information overload scale items were summed to a one-factor solution to align with the design of the original CIO score. 

Qualitative study

More information is needed to enable the reader to evaluate the qualitative study data. It would be helpful to have a brief reference to the authors’ epistemological stance along with a statement of reflexivity which is helpful to understand the relationship of the researchers to the data collection and insight generation process.

Authors’ Response: We have added in the following paragraph “Analyst triangulation was used during the coding process to improve the reliability of the data, acknowledging that each researcher brings their own attitudes, values and world views to their interpretation of the interview transcripts. Incorporating multiple researcher viewpoints and skillsets into the analysis mitigates the risk of bias or favoring a particular point of view. Prior to analysis, each researcher also reflected on their pre-existing perspectives, assumptions and expectations of the data in order to identify implicit biases, thus further improving the reliability of the data.”

More information is needed about the sampling strategy. 35.6% (n=136) agreed to be interviewed, it would be good to know what the strategy was to decide who to contact – was a purposive or consecutive sampling approach taken and with what justification. 

Authors’ Response: Recruitment was consecutive convenience sampling based on staff availability. Families were invited whenever research staff was available for recruitment. When staff was available, all potentially eligible families were contacted. 

It would be great to have a more detailed description of the analytical process undertaken to develop the coding framework – including reference to whether those who completed the analysis were the same or different people to those who conducted the interviews, was coding inductive or deductive, how did you arrive at the key themes of ‘barriers’ and ‘facilitators’ and what did the barriers and facilitators relate to – information overload or general communication or something else?

Authors’ Response: Additional information has been added to the manuscript to describe the process. 

Presentation of quotes from participants should ideally be at a minimum labelled with a participant ID so that the reader can be reassured that the quote come from a number of participants. It can also be useful to have other descriptives e.g. gender, or education level so that there is some contextual information to help interpret the quotes offered.

Authors’ Response: We have added in a “number” column for each subcode to ensure that readers are aware that subcodes were discussed by multiple participants. We have included the participant ID after each quote. Quotes are included to exemplify the key themes identified, rather then explore an individual patients perspective. We therefore believe descriptive or contextual information for each quote is not required. General demographic information is included for those who completed the qualitative interview stage. 

Relationship between the quantitative and qualitative study

The two studies feel unconnected at the moment, and I wonder if more could be done to connect the findings of the two studies. The qualitative study appears to be a more general description of communication experiences of parents rather than a more in-depth exploration of the experience of information overload. This may reflect the low prevalence of information sample in the questionnaire study from which the qualitative sample was drawn. Purposive sampling to specifically invite questionnaire participants with higher CIO scores to the interview study may have enabled this in-depth exploration and I think this is a potentially limitation of the paper as it is presented.

Authors’ Response: Thank you for your interesting observation. We did not do purposive sampling. It may have been interesting to speak with those with higher information overload however, we did not want to introduce selection bias but rather gain a more general understanding among our patient population. 

Below are a few additional points that I hope you might find helpful. I found that in places the manuscript was not logically organised (e.g. description of all the variables used in the quant study).

Introduction

Acknowledging the comment that there is little specific research that has been done to measure information burden in parents whose child has under gone elective surgery, I wonder if there may still be benefit to adding more contextual research that could give the reader some ideas about what factors might be influential here. E.g. Potential parental demographic, psychological variables, child medical details, HCP related factors. They are hinted at but not referenced e.g. line 99 and 100 or referenced in the discussion, line 397 Khaleel reference about association of SES with information overload.

Authors’ Response: We respectfully feel that these points are adequately outlined in the introduction and discussed where relevant to our findings in the Discussion. 

Methods

Line 180 recommend use of “variables” rather than “factors” as it is confusing to refer to factors straight after mentioning EFA.

Authors’ Response: Thank you, we have made the suggested change.

It would add to clarity if there was a specific participants/ procedure section and separate measures section. Currently reference to some of the measures only appear in the analyses/ results e.g. prior experience of health care and language spoken at home – with the abbreviation LOTE referenced without introduction as far as I can see. A copy of the questionnaire could be added as a supplementary document for additional information.

Authors’ Response: We have created a third supplement (Supplement 3) containing a full list of the questions used in the day-of-surgery survey. 

Results:

Regression analysis – description feels out of order by starting with reference to the interaction results – clarity / logical order would be improved if started with reference to the unadjusted results of all the variables (table I think is in supplementary) and go from there, building up to the interaction results. I haven't unfortunately had time to consider the statistical analysis in detail so I have answered "Don't know" to the question related that asks me to assess whether the statistical analysis has been completed appropriately and rigorously.

Authors’ Response: In statistical modelling, the unadjusted estimates can be misleading and in fact be entirely incorrect. Adjusted estimates allow us to isolate the effect of the key variable(s) of interest after adjusting for effects of confounding variables. Given the comments from yourself and Reviewer 1, we have included a table of the unadjusted results as a table in Supplement 2. However, we disagree that the unadjusted single-variable regression estimates should be reported before the adjusted estimates in the main text, given the caveats that unadjusted estimates can be misleading. We have added a sentence referencing that the unadjusted estimates are available in Supplement 2 to the results, but we disagree about altering the order in which the results are presented given these concerns.

The following sentence has been added to the regression analysis results:

Unadjusted regression estimates from single-variable models for total score on each candidate independent variable are reported in S2 Table 1. However, these do not reflect the true effect of each variable as this analysis does not account for the effect of confounding variables. 

Title of table 6 could be revised for clarity – I would remove reference to stage 3 and add something that tells the reader explicitly what the barriers and enablers specifically relate to e.g. general communication or avoidance of information overload?

Authors’ Response: The term stage 3 has been removed as suggested. We have adjusted the title to now read “The key themes, barriers and enablers surrounding the information received by parents regarding their child’s surgery, identified from thematic analysis of semi-structured interview transcripts “. The column headings have been revised to provide a more specific definition of the content addressed in each row.

It would be great to see themes evidenced with quotes alongside in the table

Authors’ Response: We believe that this information will overload the table and distract from the summarised results. We have included quotes alongside the themes in the text. 

I think some of the sub codes could be revised for greater conceptual clarity:

e.g. Own Research or Experience - seems to relate to two different things: 1) conflict/ confusion arising from parents finding information from the internet which is different from what they have been told by hospital staff and 2) problems related to individual characteristics of parents e.g. ADHD.

e.g. Barriers: poor communication- appears to give a description of HCP communication that was perceived to be ineffective whereas when it comes to Enablers: “good” HCP communication is sub-divided to into different categories e.g. written, verbal, rapport.

Authors’ Response: We thank the reviewer for their comments. The first subcode mentioned

---

## [Decision Letter · Decision Letter 3]

25 Jun 2024

PONE-D-23-36082R3Information overload and parental perspectives on information provided to parents/carers of paediatric patients undergoing elective surgical procedures.PLOS ONE

Dear Dr. Sommerfield,

Thank you for submitting your manuscript to PLOS ONE. After careful consideration, we feel that it has merit but does not fully meet PLOS ONE’s publication criteria as it currently stands. Therefore, we invite you to submit a revised version of the manuscript that addresses the points raised during the review process.

Please address Reviewer #1's comments, as shown below.

We look forward to receiving your revised manuscript.

Kind regards,

Boyen Huang, DDS, MHA, PhD

Academic Editor

PLOS ONE

Journal Requirements:

Reviewers' comments:

Reviewer's Responses to Questions

**Comments to the Author**

1. If the authors have adequately addressed your comments raised in a previous round of review and you feel that this manuscript is now acceptable for publication, you may indicate that here to bypass the “Comments to the Author” section, enter your conflict of interest statement in the “Confidential to Editor” section, and submit your "Accept" recommendation.

Reviewer #1: (No Response)

Reviewer #4: All comments have been addressed

2. Is the manuscript technically sound, and do the data support the conclusions?

Reviewer #1: Yes

Reviewer #4: Yes

3. Has the statistical analysis been performed appropriately and rigorously? 

Reviewer #1: Yes

Reviewer #4: Yes

4. Have the authors made all data underlying the findings in their manuscript fully available?

Reviewer #1: Yes

Reviewer #4: Yes

5. Is the manuscript presented in an intelligible fashion and written in standard English?

Reviewer #1: Yes

Reviewer #4: Yes

6. Review Comments to the Author

Reviewer #1: Comments are limited to the quantitative part of the paper:

The regression results are now easier to follow, and transparency regarding confounding factors have improved.

Some suggestions:

In the discussion, detailed results (estimates and p-values) are reported on the estimated impact of education on IOC among those not identifying as Aboriginal or Torres Strait Islander. While these figures are reported in a table in the results, the authors may choose to rather highlight them under results and give a more general description in the discussion.

In the first paragraph of the discussion there is a referral to figure 1. This can be deleted.

In the discussion, while results on education are in line with those reported in the scoping review by Khaleel et al. (2020), some are not (socio-economic factors). Is it possible to elucidate any reasons as to why?

In S2 table 1 some p-values are reported as 0. These should be reported with more accuracy, e.g as <0.001.

Reviewer #4: Manuscript Number: PONE-D-23-36082R3

Manuscript Title: Information Overload and Parental Perspectives on Information Provided to Parents/Carers of Pediatric Patients Undergoing Elective Surgical Procedures

I had the pleasure of reviewing the manuscript titled, “Information Overload and Parental Perspectives on Information Provided to Parents/Carers of Pediatric Patients Undergoing Elective Surgical Procedures.”

The manuscript has been revised based on comments and suggestions from three previous reviews. I have examined all reviewer comments and the authors' responses. The authors have appropriately addressed all requests and updated the manuscript accordingly.

I have two minor suggestions regarding lines 209 and 212: the word "predict" should be changed to the past tense "predicted."

Overall, this is a well-written manuscript that addresses an important issue.

7. PLOS authors have the option to publish the peer review history of their article (what does this mean?). If published, this will include your full peer review and any attached files.

Reviewer #1: No

Reviewer #4: No

---

## [Author Response · Author response to Decision Letter 3]

28 Jul 2024

Below are our responses to the points raised by reviewers. 

In the discussion, detailed results (estimates and p-values) are reported on the estimated impact of education on IOC among those not identifying as Aboriginal or Torres Strait Islander. While these figures are reported in a table in the results, the authors may choose to rather highlight them under results and give a more general description in the discussion.

Response: We have revised the manuscript as suggested. 

In the first paragraph of the discussion there is a referral to figure 1. This can be deleted.

Response: This has been deleted. 

In the discussion, while results on education are in line with those reported in the scoping review by Khaleel et al. (2020), some are not (socio-economic factors). Is it possible to elucidate any reasons as to why?

Response: The scoping review by Khaleel et al. reported that low education level and low socioeconomic status were associated with information overload in 3 and 2 studies, respectively. We have added to the Discussion section to expand on this. 

In S2 table 1 some p-values are reported as 0. These should be reported with more accuracy, e.g as <0.001.

Response: The suggested change has been made to Supplement 2 Table 1.

I have two minor suggestions regarding lines 209 and 212: the word "predict" should be changed to the past tense "predicted."

Response: This change has been made as suggested in the revised manuscript.

---

## [Editor Report · Decision Letter 4]

30 Jul 2024

Information overload and parental perspectives on information provided to parents/carers of paediatric patients undergoing elective surgical procedures.

PONE-D-23-36082R4

Dear Dr. Sommerfield,

We’re pleased to inform you that your manuscript has been judged scientifically suitable for publication and will be formally accepted for publication once it meets all outstanding technical requirements.

Kind regards,

Boyen Huang, DDS, MHA, PhD

Academic Editor

PLOS ONE
---

## [Editor Report · Acceptance letter]

22 Aug 2024

PONE-D-23-36082R4 

PLOS ONE

Dear Dr. Sommerfield, 

I'm pleased to inform you that your manuscript has been deemed suitable for publication in PLOS ONE. Congratulations! Your manuscript is now being handed over to our production team.

Kind regards, 

on behalf of

Dr Boyen Huang 

Academic Editor

PLOS ONE